# Structural insights into the transporting and catalyzing mechanism of DltB in LTA D-alanylation

Pingfeng Zhang [1,3] ✉ & Zheng Liu [2,3] ✉

DltB, a model member of the Membrane-Bound O-AcylTransferase (MBOAT) superfamily, plays a crucial role in D-alanylation of the lipoteichoic acid (LTA), a significant component of the cell wall of gram-positive bacteria. This process stabilizes the cell wall structure, influences bacterial virulence, and modulates the host immune response. Despite its significance, the role of DltB is not well understood. Through biochemical analysis and cryo-EM imaging, we discover that *Streptococcus thermophilus* DltB forms a homo-tetramer on the cell membrane. We further visualize DltB in an apo form, in complex with DltC, and in complex with its inhibitor amsacrine (m-AMSA). Each tetramer features a central hole. The C-tunnel of each protomer faces the intratetramer interface and provides access to the periphery membrane. Each protomer binds a DltC without changing the tetrameric organization. A phosphatidylglycerol (PG) molecule in the substrate-binding site may serve as an LTA carrier. The inhibitor m-AMSA bound to the L-tunnel of each protomer blocks the active site. The tetrameric organization of DltB provides a scaffold for catalyzing D-alanyl transfer and regulating the channel opening and closing. Our findings unveil DltB's dual function in the D-alanylation pathway, and provide insight for targeting DltB as a anti-virulence antibiotic.

The MBOAT superfamily comprises a diverse group of integral transmembrane proteins found in all kingdoms of life, which share low sequence similarity and perform divergent physiological functions[1]. In Gram-positive bacteria, the bacterial MBOAT DltB catalyzes the D-alanylation of membrane-anchored LTA, a major component of the cell wall of these bacteria[2,3]. In addition to its biosynthesis, LTA modification is essential for bacterial adaptation to the environment and is crucial for virulence and pathogenesis[4–7], it also contains immunomodulatory properties to modulate host immune response[8]. The D-alanylation process, regulated by the Dlt-operon consisting of 5 genes (named DltA-D, and an additional DltE or DltX)[2,9–11], has been proposed as a potential target for treating bacterial infections[12–15]. However, the detailed mechanism following formation of D-Ala-DltC remains poorly understood, mainly due to the unclear function of DltB,

which has been identified as a target for novel antibiotic development[13].

DltB serves as a structural model of the MBOAT family members, as its crystal structure was first determined[16] and was used for comparative analysis with successively solved structures of several other MBOAT members, including Sterol O-acyltransferase 1 (SOAT1, also known as Acyl-coenzyme A: cholesterol acyltransferase 1 (ACAT1))[17,18], Diacylglycerol O-acyltransferase 1 (DGAT1)[19,20], Protein-cysteine N-palmitoyltransferase (also known as Hedgehog Acyltransferase HHAT)[21,22], and Protein-serine O-palmitoleoyltransferase porcupine (PORCN)[23]. Despite the diversity, a universal MBOAT fold consisting of 8 TMHs exists in all these members[19,24]. Studies have revealed that some MBOAT members have dual functions, such as HHAT, which acts as a transporter to promote palmitoyl-CoA uptake across ER

[1]Cancer Center, Renmin Hospital of Wuhan University, Wuhan, China. [2]Kobilka Institute of Innovative Drug Discovery, School of Medicine, The Chinese University of Hong Kong, Shenzhen, Shenzhen, China. [3]These authors contributed equally: Pingfeng Zhang, Zheng Liu.
✉e-mail: pingfeng.zhang@whu.edu.cn; liuzheng@cuhk.edu.cn

membrane in addition to its acyltransferase function catalyzing the palmitoylation of SHH-N[25]. DltB has been proposed to serve as both an enzyme and a transporter or channel[9,26], but its actual function in D-alanylation is not fully understood.

Unlike wall teichoic acid (WTA), which is biosynthesized in the cytoplasm and then flipped outward, LTA is synthesized directly on the outside of the cells[27–29]. However, the original D-alanylation substrate, D-alanine, is biosynthesized in the cytoplasm of the cell[30]. That means the substrate has to be transported to the outside of the cell for LTA D-alanylation. DltA ligases D-alanine onto the small carrier protein DltC at the modified Ser residue (Ser35 on *Streptococcus thermophilus* DltC, which has 79 amino acids in sequence). Since DltC does not transverse the membrane, but LTA D-alanylation occurs on the outer surface of the cell[31], it is still unclear how the D-alanyl substrate is transferred cross the membrane to the outside of the cell. DltB is the only integral transmembrane protein in Dlt-operon that plays a key role in D-alanylation. In addition to its acyltransferase role, it is highly possible that it also functions as a transporter for the D-alanyl intermediate.

In this study, we set out to study the structure and function of DltB to uncover the mechanism in D-alanine incorporation. Our results shed light on targeting DltB for the development of anti-virulence antibiotics and provide a dual functional model for the MBOAT superfamily in general.

## Results

### Tetrameric DltB is the natural functional unit on the cell membrane

DltB has been proposed as the channel responsible for transporting the substrate from the cytosol to the outside of the cell for D-alanylation, as it is the only integral membrane protein in the Dlt operon[9,26]. In its monomeric crystal structure, a C-tunnel (cytoplasmic side) has been found to bind D-Ala-DltC[16] (Supplementary Fig. 1). It is uncertain if the monomer is the functional unit capable of opening to transfer the D-alanyl substrate. Studies on other MBOAT members have shown that DGAT1 and ACAT1 form oligomers, including dimers and tetramers, which are critical for their functions[17–20] (Supplementary Fig. 2). The natural oligomeric state of DltB is currently unknown.

In the process of the purification, DltB behaves as an oligomer, tetramer, dimer and monomer on a gel filtration column (Fig. 1a). The tetramer is stable when reloaded onto the gel filtration column (Supplementary Fig. 3a). Crosslinking with bis(sulfosuccinimidyl) suberate (BS3) or glutaraldehyde on purified DltB tetramers forms higher molecular weight oligomers including dimers, tetramers and even higher oligomeric species other than monomers according to the SDS-PAGE, but no higher oligomers observed for DltB monomers in the same setting (Fig. 1b, Supplementary Fig. 3b). To study the native status of DltB on the cell membrane, a crosslinking experiment was performed on isolated cell membrane or whole *E. coli* cells expressing DltB. With increasing BS3 crosslinking time, more monomers were crosslinked to form dimers, tetramers, or higher oligomers (Fig. 1c, Supplementary Fig. 3c), consistent with the result of gel filtration chromatography and the purified DltB tetramer crosslinking experiment. All these results indicate that DltB form tetramers on the cell membrane with a molecular weight of about 200 kD. To investigate what fraction of DltB form tetramers on the membrane, cell membrane containing overexpressed DltB was mildly solubilized in n-Dodecyl-β-Maltoside (β-DDM) and loaded onto a Superdex 200 column. The result demonstrated that most DltB on the membrane forms tetramers (Fig. 1d). Furthermore, negative-stain TEM observation of the purified tetramer fractions displayed well-separated homogeneous disk-shape particles about 14 nm in diameter (Fig. 1e), with some bar-shaped particles representing the stand-up view of the tetramers, at the concentration of about 200 nM (Fig. 1e, white arrows). The homogeneity of the particles confirmed the stability of the tetramer, suggesting they are the natural functional unit on the cell membrane (Fig. 1a–e).

The natural DltB tetramers tend to be disturbed by detergent during purification, they aggregate or dissociate due to variations among batches. Typically, a mixture of oligomers, tetramers, dimers, and monomers is obtained (Supplementary Fig. 3d). The use of more detergent (higher detergent-to-lipids ratio of that batch) can cause the loss of lipids bound to the DltB protein, thus affecting the oligomeric status during purification.

### Cryo-EM structure determination for the DltB tetramers

To understand the architecture of the disk-shape DltB tetramer, we continued to use cryo-EM to visualize the structure of the DltB tetramer (Supplementary Fig. 4). The particles are readily identified in the raw images as 14 nm bar-shapes that represent an upstanding tetramer disk (Fig. 1f). Most of the particles stack closely at high concentration (~10 mg·ml⁻¹), while a few particles lay down in the plane of the grid as a separate disk about 14 nm in diameter (Fig. 1f, white arrows). This further confirms a stable DltB tetramer in the purified form. 2D classification clearly shows a disk of tetramer formed of 4 protomers (Supplementary Fig. 4).

We further determined the cryo-EM structure of the tetramer form of apo DltB at 3.42 Å resolution (Fig. 1g, Supplementary Fig. 4), and DltB complexed with DltC at 3.50 Å resolution (Supplementary Fig. 5), and in complex with an inhibitor m-AMSA at 3.23 Å resolution (Supplementary Fig. 6). The cryo-EM structure reveals that the DltB tetramer forms a disk of about 40 Å thick, while the intracellular surface of the tetramer has a longer side of about 130 Å, and the extracellular surface side is about 100 Å (Fig. 1g). The local resolution distributions of the best DltB/AMSA map demonstrate that the core of the transmembrane part has better quality, at 3.1 Å resolution, while the loops extended out of the membrane have a resolution of around 3.9 Å due to flexibility (Supplementary Fig. 6, Supplementary Table 1). The high quality of the density map allows for de novo model building of the entire molecule chain, from residue Met1 to residue Lys415. In addition, several critical lipids in the substrate binding site or on the tetramer interfaces, as well as many β-DDM molecules on the surface of DltB, were identified.

### Architecture of the tetrameric DltB

Significantly different from the monomeric crystal structure of DltB[16], our cryo-EM structures reveal the tetrameric organization of DltB. Each DltB protomer in the tetramer is almost identical to the crystal structure of DltB monomer, with a root mean squared deviation (r.m.s.d.) value of 0.4 Å. DltB is composed of a conserved MBOAT fold[19] (TMH 3–10) and two additional N-terminal transmembrane helices (TMH 1 and 2) (Fig. 2a). TMH 3-5 forms a short helix bundle, named bundle 1, that is located on the outside of the tetramer. This bundle connects to bundle 2, comprising of two long helices TMH 6–7, via two short helices, and the long bundle 2 is located on the intratetramer interface and extrudes to the extracellular surface of the membrane. The helix bundle 3 (TMH 8–10) connects to TMH7 via a long loop which contains a long cytoplasmic helix (H$_C$) lying in the membrane plane and two short helices. TMH9 and 10 extrude out into the intracellular surface in the corner of the tetramer disk (Fig. 2a, b). In the image processing, application of C4 symmetry to the refinement and reconstruction did not yield better resolution map, indicating that the DltB tetramer adopts a non-4-fold symmetry. This observation is also consistent with the following structural analysis: Four Phe36 residues of the four protomers form an approximate rectangle on the intracellular side, while the four Tyr22 residues on the extracellular side are more twisted (Fig. 2c). Thus, the tetramer is a dimer of dimers, the two dimers (MolAB and MolCD) are identical.

Of note, a central hole is formed in the tetramer, with the two closer protomers about 12 Å apart (according to Phe18 on the extracellular surface), while the distance between the other two protomers is about 24 Å away (according to Tyr22 on the extracellular surface)

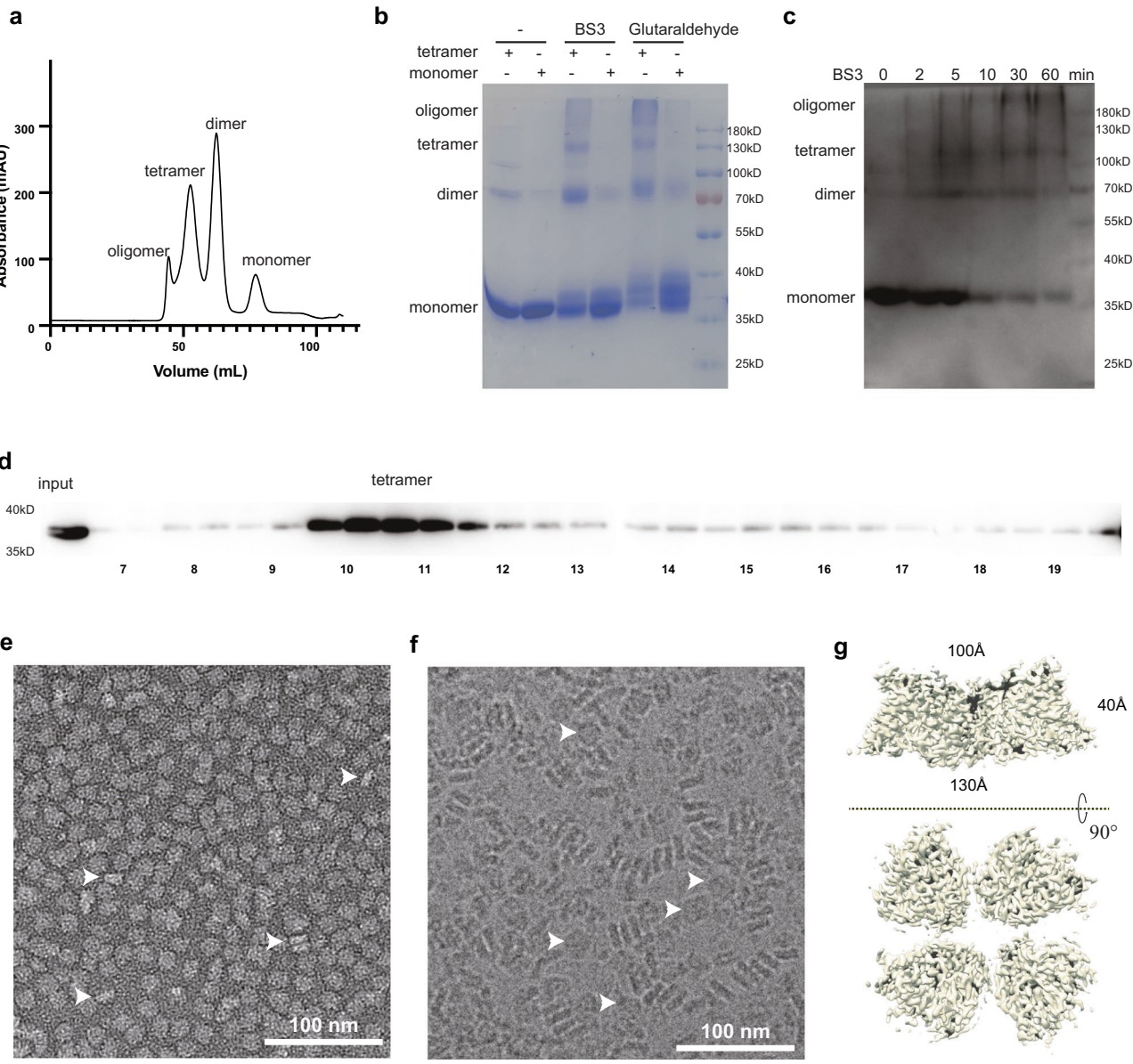

**Fig. 1 | Tetrameric organization of DltB in purified form and in cells. a** Typical gel filtration chromatography of DltB on a Superdex 200 column. **b** Crosslinking of purified DltB tetramers and monomers by BS3 or glutaraldehyde. Monomers, dimers, tetramers, and oligomers are labeled on Coomassie-stained SDS-PAGE gel. **c** Crosslinking of DltB in cells. DltB monomers, dimers, tetramers, and oligomers are detected via an anti-His antibody. **d** Isolated DltB over-expressing membrane was solubilized with 0.25% β-DDM and run on a superdex 200 gel filtration column.

Continuous fractions were analyzed on SDS-PAGE and detected by an anti-His antibody. **e** Typical negative staining image of purified DltB tetramers. **f** Typical raw cryo-EM image of purified DltB tetramers. Data represents the average of more than three independent experiments in (**b**–**f**). **g** Tetrameric DltB particle reconstituted from cryo-EM single particle analysis. Two views of the particle are presented, and the size of the particle is labeled.

(Fig. 2c). The entrance of the hole on the extracellular surface is 10 Å × 18 Å (according to Tyr22 sitting on the top), while on the intracellular surface is about 9 Å × 16 Å (according to Phe36 at the bottom) (Fig. 2c). The central hole is surrounded by the transmembrane helix TMH1 from four protomers, which is almost perpendicular to the membrane, with hydrophobic residues Phe18, Ile21, Ile25, Leu28, Ile32, Phe36 facing into the center of the hole (Fig. 2d). The four intratetramer interfaces shape up the edge of the central hole. TMH1 is the shortest helix on the extracellular surface, thus each protomer forms a funnel on the outer surface, and the four funnels further converge to the central hole. The four N-terminal helices (H$_N$) sit on the top of the intratetramer interface to build up a "wall" surrounding the central hole (Fig. 2e). The outer surface funnel has been proposed to bind the substrate LTA[16], how this organization contributes to LTA binding and

D-alanylation remains unclear. 10 DDM molecules were identified to attach to the DltB protein on the extracellular side around the central hole; in addition, an array of 2 × 5 cylindrical density covered the extracellular entrance. Since these densities are short and incomplete, no models were built on this side; however, these densities are assumed to be DDM or lipids. On the contrary, a 4 by 4 array of long cylindrical densities was built as DDM or PG on the intracellular side, and two additional ones attached in the corner (Fig. 2f). However, the biological relevance of the central hole in the tetramer needs to be further determined.

## The intratetramer interfaces inside the DltB

The intratetramer interface is mediated by the interactions between two transmembrane helices TMH2 and TMH6, while H$_N$ and TMH6

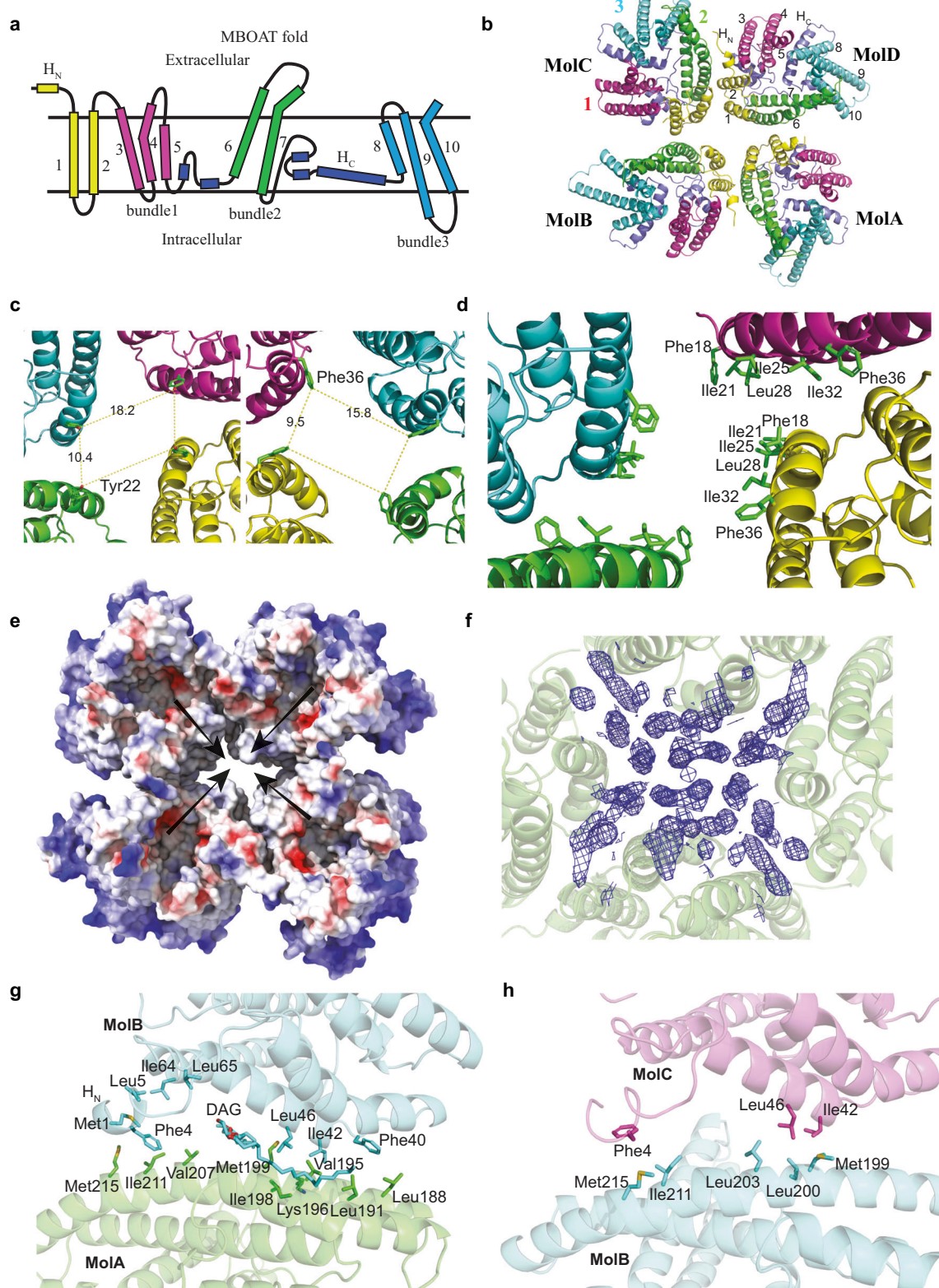

**Fig. 2 | Architecture of DltB tetramer. a** Topology model of DltB showing the 3 helix bundles. The N-terminal helix and TMH1/2 are shown in yellow, bundle 1 in red, bundle 2 in green, and bundle 3 in blue. **b** Top view of the DltB tetramer presented in a cartoon diagram with the protomers and helices labeled. The helices and bundles are color-coded as in (**a**). **c** C2 symmetry of the DltB tetramer. Two reference residues on both sides of the DltB tetramer particle are presented. **d** The surface in the central hole. The hydrophobic residues on TMH1 are highlighted and labeled. **e** The 4 extracellular funnels on each protomer converge to the central hole, with arrows showing the convergence of the funnels. **f** Both sides of the central hole are embedded by detergent molecules in the structure. The array of cylindrical densities on the cytoplasmic side (4 × 4 plus 2 in the corner) is shown as a blue mesh. **g**, **h** The MolAB (**g**) and MolBC (**h**) interfaces. The interacting residues and lipids are highlighted and labeled.

interact on the extracellular surface. As the tetramer is a dimer of dimers, there are two different intratetramer interfaces: the intradimer interface (A-B interface between MolA and MolB or MolC and MolD) and interdimer interface (B-C interface between MolB and MolC or MolA and MolD). At the A-B interface, Leu46 on MolB is placed between Ile198 and Met199 on MolA, and Ile42 from MolB is located between Val195 and Lys196 on MolA. The core interactions in this region bring the two molecules closer to each other, thereby stabilizing a lipid bound in this region, which was interpreted as diacylglycerol (DAG) by fitting into the fork-like density (Fig. 2g, Supplementary Fig. 7a, b). This interface includes comprehensive hydrophobic interactions. In the middle region, Val195 and Met199 from MolA directly interact with hydrophobic residues Ile42, Leu46 from MolB. Additionally, on the extracellular surface, $H_N$ from MolB directly interacts with both MolB (Ile64 and Leu65) and MolA (Val207, Ile211 and Met215) (Fig. 2g), while Phe18, Pro17, Leu200, Phe206, and Val207 from MolA form a hydrophobic zone close to Ile49, Thr50, Val53, Leu54, Phe4, Leu8, and Pro9 from MolB (Supplementary Fig. 7). Furthermore, on the intracellular surface, Phe40 from MolB makes Van de Waals contacts with Leu188 and Leu191 from MolA (Fig. 2g).

For the B-C interface, the core interaction region shifts, as Leu46 on MolC is placed between Met199 and Leu200 on MolB and shifts towards extracellular side (closer to Leu203 on MolB), and Ile42 from MolC moves away from Val195 but is located between Lys196 and Leu200 on MolB (Fig. 2h). This shift in the core region causes movement of the whole molecule, and no obvious lipid density is identified in this region. The general interface is still similar, which is mediated by TMH2 and TMH6, but helix $H_N$ from MolC is a little further away from MolB, and Phe40 on MolB is much further away from Leu188 and Leu191 on the intracellular surface (Fig. 2g).

To explore the intratetramer interface for the tetrameric assembly, key residues Ile42, Leu46, and Leu200 on the intratetramer interface were mutated to arginine, and Met199 was mutated to alanine, respectively, to potentially induce DltB into a dimer or monomer. After mild solubilization of the membrane containing each mutant protein with 0.25% β-DDM, the protein samples were loaded onto a Superdex 200 column to monitor the aggregation status by western blotting of the fractions. While the WT DltB protein maintained tetrameric organization, the four mutations on the tetramer interface impaired the tetramer organization on membrane (Supplementary Fig. 8). Our results suggest that the tetramer interface mutations render the tetramer vulnerable.

## DltC delivers D-alanyl to DltB without changing the tetramer conformation

D-alanylation of LTA involves a series of biochemical reactions with five Dlt-operon members. DltC is the D-alanyl carrier protein, which undergoes posttranslational modification by acyl carrier protein synthase (AcpS) to attach a 4'-phosphopantetheine (Ppant) group to a conserved serine residue (Ser35 on DltC-ST). Following D-alanine is activated to form D-Ala-AMP in the cytoplasm, the D-alanyl group is loaded onto the thiol of Ppant on DltC by the ligase DltA[32,33]. Ppant-DltC forms a stable complex with DltB, as evidenced by the crystal structure of the DltB/Ppant-DltC complex in a monomer form, showing that DltC carries the D-alanyl group onto the intracellular surface of DltB[16]. To understand how the D-alanyl group is delivered to the DltB tetramer, we determined the tetrameric structure of DltB/DltC complex. Even without the Ppant modification on DltC, this binding binds to the intracellular surface of DltB. Interestingly, it does not affect the organization of the DltB tetramer, as the DltB tetramer exhibits an r.m.s.d. value of 0.5 Å when compared to the apo DltB tetramer (Fig. 3a). Furthermore, the tetramer also remains unchanged in the DltB/C complex, with an r.m.s.d. value of approximately 0.58 Å between our DltB/C monomer and the DltB/C crystal structure.

DltC binds to the intracellular side of DltB via hydrophobic interactions (Met36 and Val39 from DltC interact with Met302, Val305, Ile306, and Met309 on the horizontal helix $H_C$ on DltB) and salt bridges (Glu40 from DltC interacts with Lys312 on the tip of $H_C$ and Arg317 on TMH8 from DltB). It sits on the $H_C$ helix at the entrance of the C-tunnel (Fig. 3b). Ser35 on DltC is approximately 20.4 Å away in straight distance from His336 residue at the active site on DltB. With the Ppant group attached to Ser35, it can insert into the tunnel with an extended conformation that reaches out to about 20 Å (Supplementary Fig. 9). Thus, considering the distance, DltC can bring the D-alanyl group to the active site of DltB directly. However, the tunnel is blocked by Trp285, Met332, and Ser293 in the current structures (Supplementary Fig. 9). Sterically, it is not possible for DltC to deliver the D-alanyl group to the active site of DltB in the resting state. This means a conformational change in DltB is required for DltC to deliver the D-alanyl group to the active site. Helix $H_C$, which connects helix bundles 2 and 3, may play a critical role in the conformational change. As helix bundle 3 is located at the corner of the DltB tetramer, it is possible for bundle 3 to move away from bundle 2, where the long loop containing the $H_C$ helix is involved in mediating the conformational change. This will be further discussed in the next section.

## LTA substrate binding site on DltB

Intriguingly, a hydrophobic cleft is formed between helix bundle 1 and 3; furthermore, a continuous density is obviously identified between TMH4 (from bundle 1) and TMH8 (from bundle 3), sitting on the horizontal cytoplasmic helix $H_C$ in the membrane (Fig. 3c). As this density exists in all DltB tetramer structures that were determined in this study, we concluded that it is derived from the purified DltB protein, and probably it is a lipid from the host cell membrane. This density does not exist in the crystal structure of DltB (Supplementary Fig. 10), which may be lost during purification or be flexible in the crystal. This cleft is basically hydrophobic, surrounded by Leu304, Val307, Ile327, Val331, Phe334, Ile338, Val110, Ile106, Leu105, Val102, Tyr95, Phe94, Ile138, Val134, Phe131, and Trp295 (Fig. 3d). Considering the DltB substrate LTA has to be anchored onto the membrane, it is highly possible that the lipid bound here serves as a carrier for LTA, hooking LTA to the outside of DltB.

LTA is a hydrophilic chain of variable repeating units, typically polyglycerol-phosphate, covalently attached to the head group of an anchor glycolipid, predominantly diglucosyl-diacylglycerol (Glc₂-DAG)[8]. DAG or PG has also been suggested as an intermediate LTA anchor during biosynthesis[29,34]. Based on the shape of the electron potential density in the cryo-EM map, a PG molecule was modeled into this density, with the glycerolphosphate head group facing the extracellular side, making contact with the main chain of residues Val110 and Ile338 (Fig. 3d, Supplementary Fig. 10). In some cases, an additional density is assigned as a DDM molecular (Fig. 3d). As this hydrophobic cleft is sterically close to the active residue His336, we propose it is the substrate binding cleft for lipid-anchored LTA.

This cleft is conserved in all MBOAT members. A continuous density can be identified in several structures. In ACAT1 structure 6vum, a cholesterol was assigned to this cleft[17]; in 6p2p, a density was identified but not assigned. In DGAT1 structure 6vp0, a tubular density was observed and proposed as DAG entrance[19]. In PORCN structure 7ura, a detergent digitonin molecule (AJP) was assigned in this cleft; in 7urd, a similar density was identified but not assigned[23] (Supplementary Fig. 11). All these indicate this cleft is conserved in MBOAT to bind some substrates or lipids.

To investigate the nature of the lipid bound in this cleft, we performed lipidomic analysis on the purified DltB tetramer and monomer samples together with the starting membrane material and crude DltB sample from the Ni column (Supplementary Table 2, Supplementary Data 1). Consistently, a significant specific accumulation of PG on the

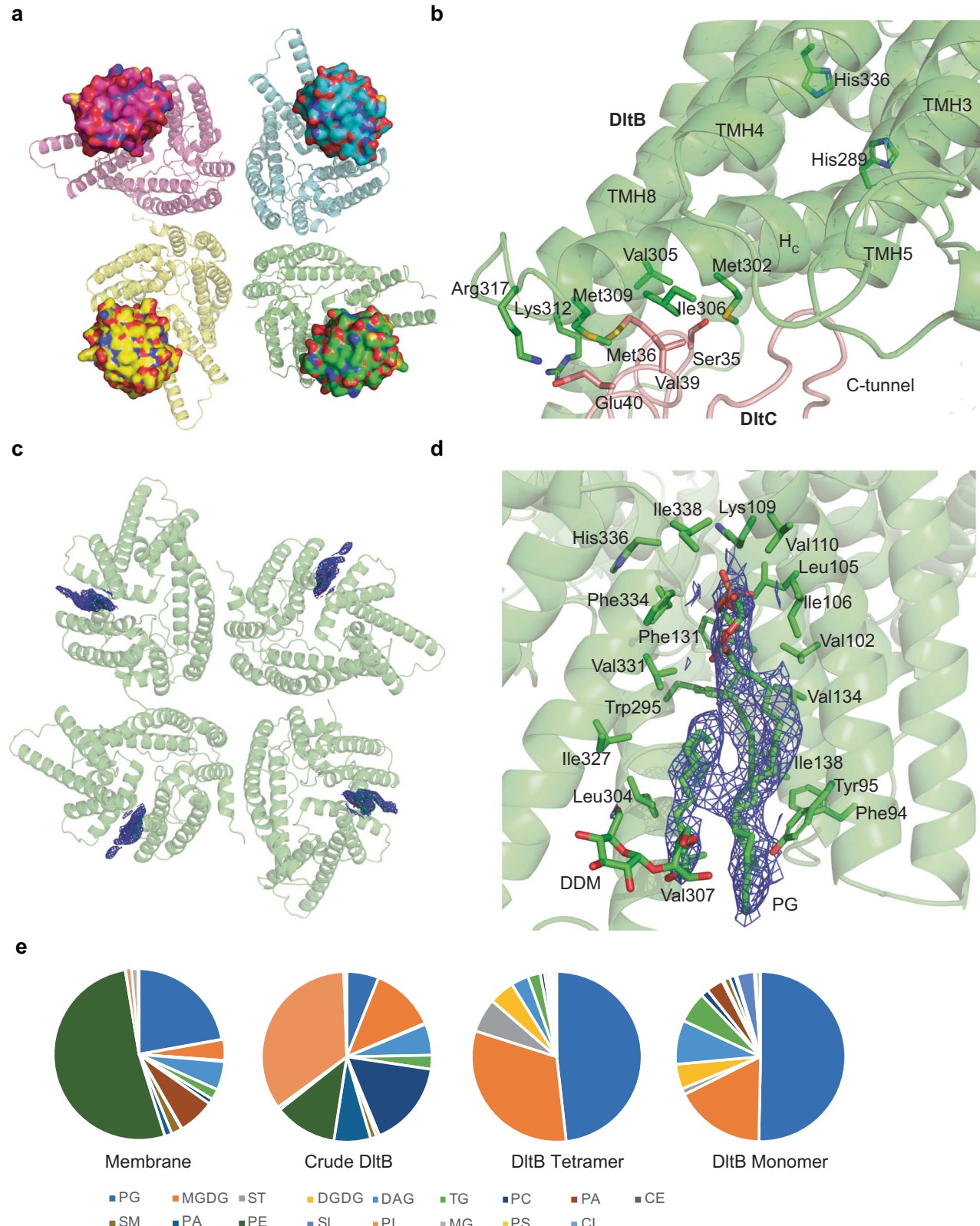

**Fig. 3 | Substrate D-alanyl-DltC and lipid-LTA binding sites on DltB. a** DltC binds to the DltB tetramer. The DltB tetramer is represented as a cartoon, with 4 proto-mers colored differently. The surfaces of the 4 DltC molecules are shown in the corresponding color of DltB. **b** The interaction network between DltB and DltC. The interacting residues are highlighted as a stick model and labeled. DltB is in green, while DltC is in red. **c** Lipid binds to the substrate binding cleft. The electron potential density map of the PG molecule is shown as a blue mesh (contoured at 5σ level). **d** The hydrophobic substrate binding environment. The side chains of the hydrophobic surrounding residues are highlighted as a stick model and labeled. **e** Pie charts representing the relative lipid composition of the membrane, crude DltB, DltB tetramer, and monomer are presented. The lipid identities are color-coded in the pie charts and ordered from most abundant to the least in the tetramer.

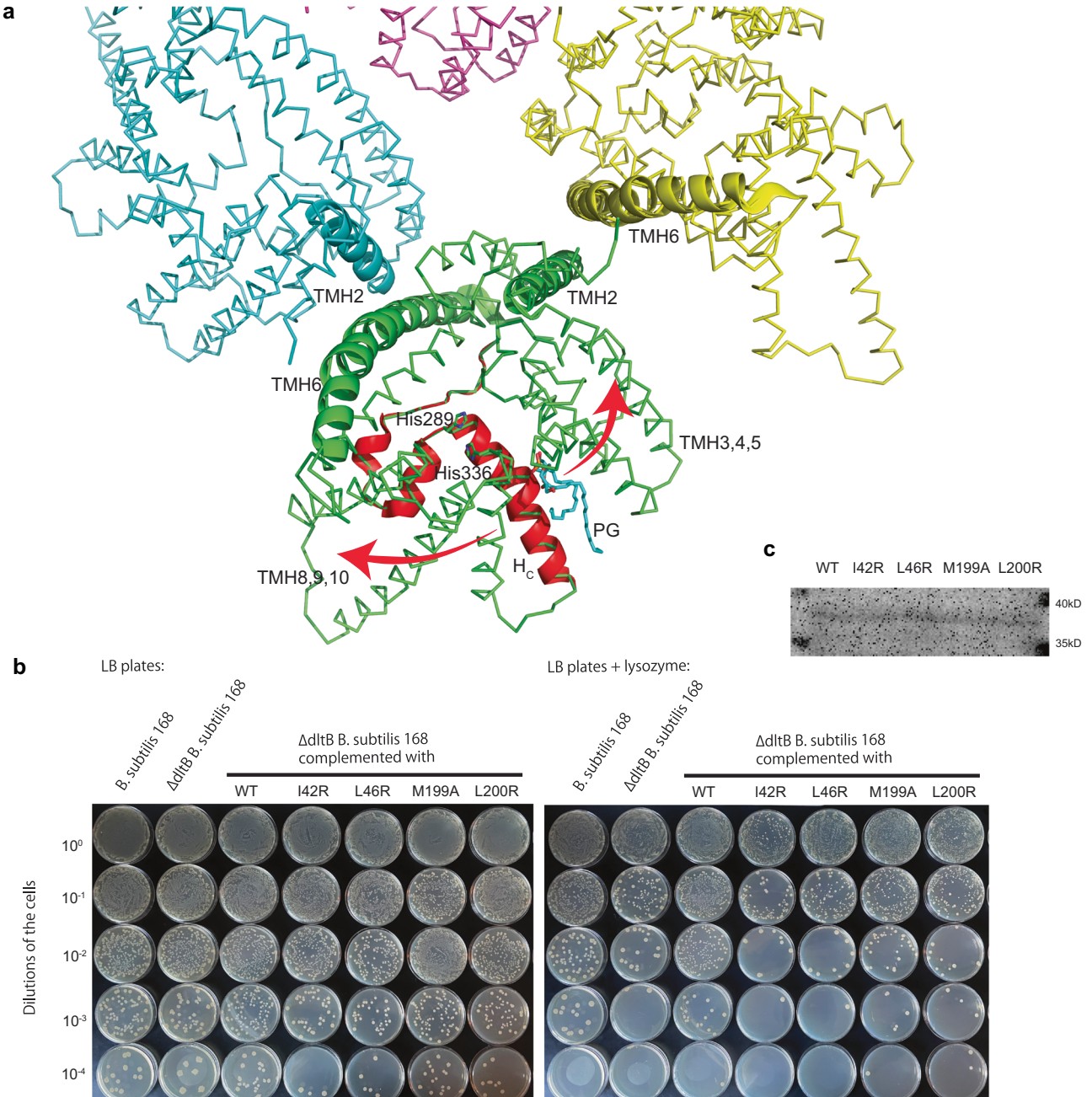

**Fig. 4 | DltB tetramer provides a Scaffold to open the channel for LTA D-alanylation. a** A working model for the DltB conformational change to open the channel in the DltB monomer. DltB molecules are presented in ribbon form, and the transmembrane helices on the interfaces are highlighted with carton models in the same color of the individual DltB molecules. The horizontal helix $H_C$ and two short helices are colored in red. The PG in the substrate cleft is shown in stick model. The directions of the conformational changes are indicated with arrows. **b** Lysozyme susceptibility survival assay with WT DltB and the tetramer interface mutants in *B. subtilis*. Representative images are shown for serial dilutions of cells carrying various DltB mutants plated on LB agar (left) and LB agar supplemented with 3 μg ml⁻¹ of lysozyme (right). Dilutions of cells are indicated on the left of the images. **c** The DltB WT and mutant proteins fusion with a C-terminal His-tag expressed in *B. subtilis* for the survival assay were detected by an anti-His antibody. The assays in (**b**, **c**) were performed three times.

DltB tetramer was observed, about half of the lipids in the purified DltB tetramer protein being PG. Moreover, it continues to concentrate in the monomer fraction, with more than 50% of the lipids being PG (Fig. 3e), suggesting it is the most stably bound lipid on DltB, confirming our model that PG bound to substrate binding cleft. In addition, lipidomic data confirmed that DAG is a major component in the purified DltB protein, indicating a strong connection between DAG and DltB, which supports our model with a DAG on the intratetramer interface.

## DltB tetramer provides a Scaffold to open the channel for LTA D-alanylation

The current structures represent a closed status of DltB, in which the cleft is separated from the extracellular active site by the side chains of Trp335 (TMH7) and Lys109 (TMH4), thus it is not connected to His336 (Fig. 3d). If a slight conformational change occurs, which causes the side chain of Lys109 to move away, the lipid carried LTA can readily approach the active site of DltB. In the DltB tetramer, this cleft faces outwards of the tetramer disk, opposite to the central hole (Fig. 3c).

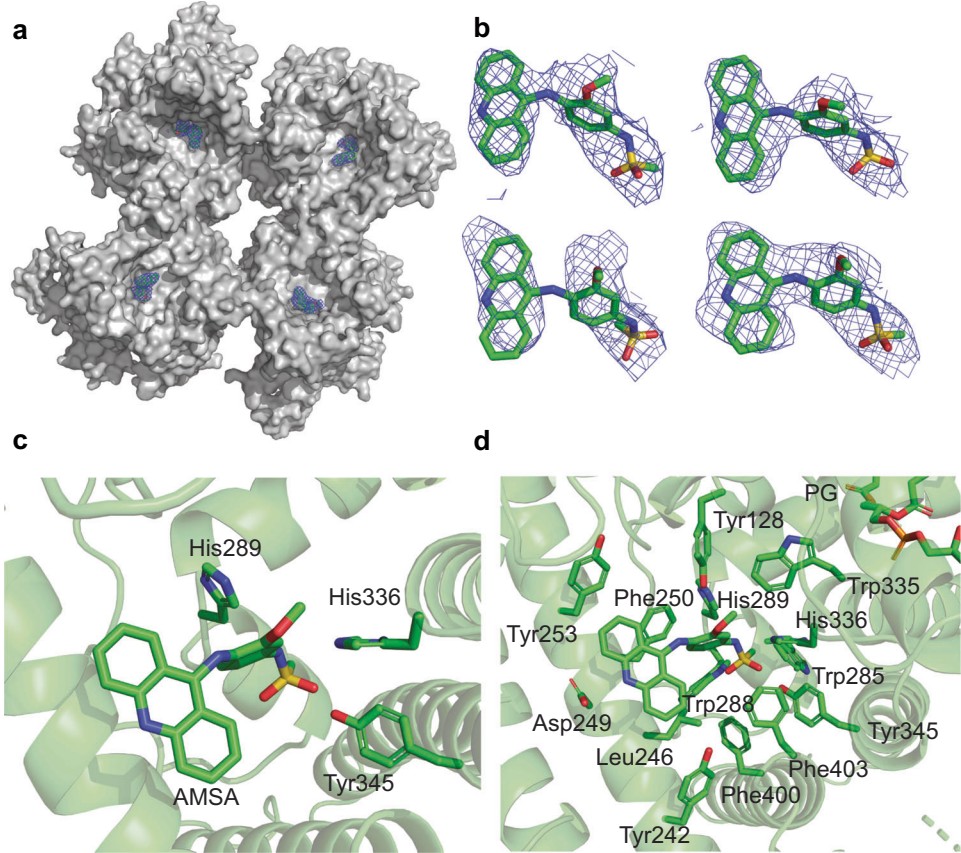

**Fig. 5 | m-AMSA binds in the L-tunnel to inhibit DltB. a** Overview of 4 m-AMSA molecules bound to each protomer. The DltB tetramer is shown as a grey surface, while the density of each m-AMSA molecule is shown in blue mesh (contoured at 5σ level). **b** Close-up view for each m-AMSA molecule fitted in the density map. The m-AMSA model is shown in green stick form, and the electron potential density map is shown in blue mesh (contoured at 5σ level). **c** m-AMSA binds to the active site of DltB. The side chains of the key residues His336, His289 and Tyr345 are shown as stick models. **d** m-AMSA binds in an environment formed by aromatic amino acids. m-AMSA and the side chains of the aromatic cluster are highlighted with stick models and labeled.

Since the tetramer is not rigid, the conformational change is feasible as follows: the two helices TMH2 and TMH6 on the tetramer interface serve as two lever fulcrums, and the free space on the tetramer interface allows the two transmembrane bundles 1 (TMH3-5) and 3 (TMH8-10) to move away from each other (Fig. 4a). Inside the DltB monomer, the two short helices and $H_C$, which sit in the membrane plane connecting TMH7 and TMH8, act as a spring to allow the two bundles to open and close (Fig. 4a).

To test this hypothesis, we generated a DltB knockout *Bacillus subtilis* 168 strain and then complemented it with the *S. thermophilus dltB* wild-type gene or various mutants on the tetramer interface. Using a lysozyme-sensitivity assay[16,35], we found that DltB deficiency increases sensitivity to lysozyme. While the WT DltB protein rescues the lysozyme vulnerability, the 4 intratetramer interface mutants failed to do so (Fig. 4b, c). Together, our findings suggest that the DltB tetramer provides a scaffold for conformational change, which occurs during LTA D-alanylation. It allows the D-alanyl group from DltC to reach the active site and LTA substrate on the outside of DltB.

### DltB inhibition by m-AMSA from extracellular surface

To understand how the DltB inhibitor m-AMSA regulates DltB activity, the structure of the DltB tetramer complexed with m-AMSA was solved by cryo-EM (Supplementary Fig. 6). The high-resolution electron potential density map (3.23 Å) of DltB/m-AMSA enables us to identify 4 m-AMSA molecules that bind to each protomer in the tetramer unambiguously, with methane-sulfonamide group deeply heading into the active cavity of DltB (Fig. 5a, b). m-AMSA binds to the extracellular surface of DltB, at the bottom of the outer surface funnel where the

active His336 is located. The sulfonamide group is embedded between the active residues His289 and His336, and forms a hydrogen bond with Tyr345 (Fig. 5c). Interestingly, the m-AMSA binding pocket is formed by an aromatic cluster, consisting of Tyr128, Trp335, His336, Phe403, Tyr345, Trp285, Trp288, Phe250, His289, Tyr253, Phe400, and Phe247 (Fig. 5d). Aromatic clusters are relevant to protein stability and determine the thermal stability of thermophilic proteins[36], they are also highly prevalent in protein-protein/protein-drug interactions[37], making m-AMSA a strong candidate for antibiotic development. In addition to this aromatic cluster, a few residues with linear side chains such as Leu246 and Asp249 are also located on the surface of the binding pocket, enhancing the interaction between m-AMSA and DltB (Fig. 5d). The binding pocket is relatively rigid, as m-AMSA doesn't significantly change the conformation when compared to the apo-structure.

Recently, the DltX protein in the Dlt operon has been found to form a complex with DltB and DltD. Its C-terminal motif (FLYFGF), which is conserved and required for LTA D-alanylation, is predicted to face the extracellular surface and bind to the active site of DltB[11]. With several aromatic residues in the DltX C-terminal motif, it is highly possible that it will bind to the DltB aromatic cluster. Thus, m-AMSA may prevent the binding of DltX.

## Discussion

### Oligomer organization for MBOAT members

Many proteins can form oligomers, but monomers are often used for structural studies. For example, the SHH receptor PTCH can be purified as both an oligomer and monomer according to size exclusion

chromatography. While most species exist as oligomers, which may be tetramers, only the monomer form was used for structural analysis, and the biological relevance of the oligomer remains unclear[38]. Therefore, there is a need to elucidate the native oligomeric form of many proteins to reveal their physiological functions. DGAT1 and ACAT1 can form dimers or tetramers that organize in different ways[17–20]. Both DGAT1 and ACAT1 dimers are mediated by TMH1 (which is not part of the MBOAT fold) and the helix bundle 2 (TMH5). Interestingly, the DltB tetramer interface is mediated by TMH2 and TMH6, corresponding to TMH1 and 5 in DGAT1 and ACAT1. TMH2 in DltB is not part of the MBOAT fold, and the additional TMH1 was placed in the central hole. Although the intratetramer interface in DltB is rather distinct in detail from the interface in other MBOAT members, the overall organization is similar: helix bundle 2 from MBOAT fold is involved in the interaction, whereas bundles 1 and 3 are free from the interaction. This organization keeps the substrate binding cleft free from intramolecular interaction, leaving it accessible to the cell membrane and feasible for conformational change. This may explain why the dimer interface mutant of ACAT1 loses its catalytic activity[18]. Our biochemical analysis and cryo-EM structure of DltB revealed its native tetramer organization on the membrane and provides a platform to uncover its role in LTA D-alanylation.

## Dual function for MBOAT members

MBOAT acts as an acyl transferase, with acyl doners usually coming from the cytosol, such as palmitoyl-CoA. Therefore, the acyl doner substrates bind to the intracellular surface, defined as the C tunnel, but extend to the extracellular surface, where the active sites are located. For example, the active site residues Asn421 (located on the tip of horizontal helix $H_C$) and His460 (located in the middle of helix TMH7) of ACAT1, the active residues Asn378 (located on the tip of horizontal helix $H_C$) and His415 (located in the middle of helix TMH7) in DGAT1, as well as Asp339 and His379 (located at the tip of TMH7) in HHAT, and Asn301 and His336 (located at the tip of TMH7) in PORCN, face the extracellular surface.

The acyl receptor usually binds to the extracellular side, such as SHH-N and Wnt, which bind to the lumen side of HHAT and PORCN, respectively. The acyl receptor from membrane usually binds via the T-tunnel (transmembrane tunnel), such as cholesterol and DAG, which bind to ACAT1 and DGAT1 in this way. Consistent with other MBOAT members, the active residues Asn289 and His336 in DltB are located on the extracellular surface (lumen, L-tunnel). DltB functions as an acyl transferase, with its D-alanyl donor, D-alanyl-DltC, binding to the intracellular side (cytosolic, C-tunnel), and the receptor LTA anchored via a lipid to the substrate binding cleft in DltB, with the hydrophilic LTA residing on the L-tunnel of DltB.

Unlike MBOAT members DGAT1 and ACAT1, which catalyze the interaction in the membrane and release the products to the membrane, the LTA D-alanylation needs to bring the acyl donor D-alanyl to the outside of the cell, making it accessible to LTA. However, D-alanyl-DltC does not transverse the membrane. Therefore, the D-alanyl substrate needs to be transferred to the outside, either via a channel or by conformational change in DltB. Up until now, the direct substrate for D-alanylation remains controversial, and how the substrate is transferred to the outside is still unknown. Several models of this DLT pathway have been proposed[2]. Our structures reveal a tetramer scaffold for the conformational change in DltB, enabling it to act as a channel to transfer the D-ala substrate, which is either directly from D-Ala-DltC or from some other D-Ala-lipid intermediate[39]. This suggests that DltB possesses dual functions as an acyltransferase and channel. While the transporter property of other MBOATs such as HHAT remains unclear[25], our DltB tetramer structure serves as a dual function model for other MBOAT family members.

The D-alanylation machinery requires a regulatory mechanism or a switch on DltB to turn the channel on and off. This means that a mechanical force or energy input needs to be applied to the DltB MBOAT fold to cause the conformational change, thus opening the DltB tunnel, making D-alanyl from DltC accessible to the active site of DltB. At the same time, the lipid-LTA needs to bind to the substrate cleft and approach the active site. We propose two possibilities in the following: First, the DltC binding site in C-tunnel is on the outside of the horizontal helix $H_C$, opposite to the substrate cleft. In the dynamic D-alanylation process, DltC delivers a D-alanyl group to the C-tunnel for the first round of reaction. For the next round, DltC leaves the DltB surface after the reaction, thus applying mechanical force on the helix $H_C$ to open the substrate cleft. Second, it is well established that all genes in the Dlt operon participate in D-alanylation, including the single-pass membrane protein DltD and DltX[11,40], which is required but its actual role in D-alanylation is unclear. A catalytic triad is essential for the enzymatic transferase function of DltD; however, this triad is separated from the membrane plane and is not proximate to the DltB active site; thus, sterically, it is not possible for DltC to deliver D-alanyl group to DltD. Considering that the enzymatic activity of DltD extracellular domain may not directly involve the LTA D-alanylation, it is possible that the transmembrane helix of DltD and DltX may associate with DltB and play a regulatory role on DltB conformational change. Indeed, a recent study demonstrated that DltD forms a stable complex with DltB, together with DltX; and the stoichiometry of DltB and DltD is probably 1:1[11]. The DltD and DltX binding site on DltB and the dynamic regulatory mechanism needs further studies.

## Antibiotic discovery

Currently, there is a multidrug-resistant infection crisis in human history. The overuse of antibiotics has resulted in the emergence of various superbugs, including several gram-positive bacteria, such as methicillin-resistant *Staphylococcus aureus* and vancomycin-resistant *Enterococcus*. The lack of new antibiotics has become a major concern for public health[41]. Most current antibiotics targeting gram-positive bacteria interfere with the cell wall, especially the peptidoglycan biosynthesis. As drug resistance continues to increase, the development of antibacterial compounds is limited due to the deficit of targets or pathways.

The DLT pathway is a perfect antivirulence target, as inhibiting D-alanylation reduces the virulence of pathogens without killing them, thus exerting no survival pressure to select more drug-resistant strains. m-AMSA has been identified as a DltB-specific inhibitor[13]. Our cryo-EM structure of DltB with m-AMSA provides a solid structural basis to target LTA D-alanylation for treating gram-positive pathogens. As an efficient DltB inhibitor[13], binding to the extracellular surface of DltB makes it accessible to pathogens. The binding of m-AMSA to the active site of DltB (close to the invariant His336 and conserved His289 in DltB) is unambiguous. It may also prevent the organization of DltBDX complex. Our structural model clearly explains how it blocks DltB activity and provides the structural basis for the development of antibiotics targeting DltB. This may pave the way for antibiotic discovery and combatting drug-resistant infections.

## Methods

### Expression and purification of DltB

The *dltB* gene from *Streptococcus thermophilus* was cloned into a pRSF vector and expressed in *E.coli* C41(DE3) cells. The cells from 12 L LB media were lysed in 50 mM Tris-HCl pH7.4, 150 mM NaCl using a French Press. The cell membrane was isolated via ultracentrifugation at 100,000 × *g* for 3 h at a low temperature. The protein was solubilized with 0.5% triton X-100 and then loaded onto Ni-NTA agarose for purification using the N-terminal hexahistidine tag affinity. Triton X-100 was replaced with 0.03% β-DDM in 50 mM Tris-HCl pH 7.4, 150 mM NaCl, 20 mM imidazole on Ni-NTA agarose, and then the DltB protein was eluted by 250 mM imidazole. Gel filtration was then

performed on a Superdex 200 column using 20 mM Hepes-NaOH pH 7.5 and 0.03% β-DDM. The tetramer fractions of DltB were concentrated to about 10 mg·mL$^{-1}$, aliquoted, and flash-frozen for further structural studies.

## Purification of DltC

*dltC* from *Streptococcus thermophilus* was cloned into a modified pET28a vector with a TEV cleavage site. DltC protein was induced in *E.coli* BL21(DE3) cells with 200 µM IPTG. Cell lysate in 50 mM Tris-HCl pH7.5 and 500 mM NaCl was loaded onto a 5 mL pre-packed Ni column and eluted with an imidazole gradient from 20 mM to 250 mM. His-tag was removed by TEV cleavage, and DltC was further purified using a Resource Q column in 20 mM Tris pH8.0 and 0-1 M NaCl. The protein was concentrated to approximately 20 mg·mL$^{-1}$, aliquoted, and flash-frozen for further use.

## Cross linking assay

Purified DltB protein (tetramers or monomers) were adjusted to about 1 mg·mL$^{-1}$, 2.5 mM BS3 or 0.01% glutaraldehyde were used to crosslink at RT for 20 min or on ice for 1 h. Crosslink of purified DltB protein was checked by Coomassie staining, and the DltB protein in cells or on membrane was visualized by western blot using an mouse anti-his antibody (Protein Tech, #66005-1, 1:2000) and HRP-linked anti-mouse IgG antibody (CST, #7076, 1:5000).

## DltB tetramer negative staining

The DltB tetramer sample was diluted to about 200 nM in buffer 20 mM Hepes-Na pH 7.5, 0.03% β-DDM. 3 µL protein sample was applied to glow-discharged (Glow machine USA) carbon-coated copper grids (EMS Formvar Carbon Film grid #8024 USA) and set down for 1 min. The extra sample was wicked away by the use of a filter paper (Whatman USA), and then the grids were stained in about 3 µL 2% uranyl acetate (pH 7.0) for 1 min. The extra buffer was wicked away by a filter paper, and the grid was dried under infrared light for about 2 min before being loaded onto the column of the Talos 120 C microscope. The microscope was operated at 120 kV, with an imaging magnification of 92,000X and a preset defocus of −2.0 µm.

## Cryo-grids preparation and data collection

Concentrated DltB tetramer sample (10 mg·mL$^{-1}$) was used to incubate with purified DltC protein (1:1 ratio), or DltB inhibitor m-AMSA (1:3 ratio) for 30 min. For cryo-EM sample grid preparation, 3 µL aliquots of the protein samples were applied onto glow-discharged holey carbon copper grids (Quantifoil Au R1.2/1.3, 300 mesh). After incubation on the grids at 4 °C under 100% humidity for 10 s, the grids were blotted for 3.0 s and then plunged frozen into liquid ethane cooled by liquid nitrogen using a Vitrobot (Mark IV, Thermo Fisher Scientific). Cryo-EM datasets were acquired on the Titan Krios microscope (Kobilka Cryo-Electron Microscopy Center, the Chinese University of Hong Kong (Shenzhen)) operating at 300 kV, equipped with a Gatan K3 Summit detector and a GIF Quantum energy filter with a 20 eV slit width. Images were recorded with SerialEM 3.80, with magnification at 105 K and a pixel size of 0.83 Å. Movie stacks were automatically acquired in super-resolution mode with 2-time hardware binning (105,000 × magnification) using SerialEM, with a defocus range from −1.0 µm to −2.0 µm. Each stack was exposed for 2.5 s with 0.05 s per frame, resulting in 50 frames and a total dose of approximately 22 e-/pixel/s.

## Image processing and cryo-EM structure determination

The general strategy in the image processing follows the method described previously[42,43] in a hierarchical way. Raw movie frames were aligned with MotionCor2 using a 9 × 7 patch, and the contrast transfer function (CTF) parameters were estimated using Gctf and CTF in JSPR[44]. Only the micrographs with consistent CTF values, including defocus and astigmatism parameter, were kept for the following image processing. Data binned by 4 times were used for micrograph screening and particle picking. The data with 2-time binning was used for particle screening and classification. The particles, after initial cleaning, were subjected to particle extraction using the originally cleaned micrograph, and the resultant dataset was used for final cleaning and reconstruction. All resolutions were estimated by applying a soft mask around the protein density, and the gold-standard FSC=0.143 criterion. ResMap[45] was used to calculate the local resolution map.

For the DltB apo form, 4769 micrographs were obtained after careful filtering by motion correction, CTF determination, and micrograph screening from 5709 movies. A total of ~5500 particles were picked by blob picking, followed by two runs of 2D classification for automatic particle selection template. Exactly 3,538,544 particles were selected based on the template picking, and 473,323 particles were obtained after several runs of 2D classification for removal of bad particles. The initial models in this study were generated by cryoSPARC ab-initio reconstruction. Relion 3D-classifcaiton sorted out 128,038 particles for the following non-uniform refinements and reconstructions in cryoSPARC. The final resolution is reconstructed to 3.42 Å (Supplementary Fig. 4 and Supplementary Table 1).

For DltB/DltC complex sample, 6018 movies were collected from the grid with the complex, and 5431 micrographs were selected after screening results of motion correction and CTF determination. A total of 2,988,656 particles were boxed with template picking function by the use of the 2D-templates generated from above processing. After several runs of 2D classification steps, 589, 450 particles were kept for the next analysis. Relion 3D classification and ab-initio reconstruction resulted in 377,918 particles for further non-uniform refinements and high-resolution reconstruction. The final construction yielded 3.50 Å cryo-EM density map (Supplementary Fig. 5 and Supplementary Table 1).

For the DltB/m-AMSA dataset, 7989 movies were obtained first, and 7549 micrographs were kept after careful filtering by motion correction and patch CTF determination. This step was followed by blob picking in cryoSPARC selection of ~9000 particles for 2D classification twice to generate an automatic picking template. After removal of bad particles and several runs of 2D classification of 4,510,317 particles, a total of 680,250 particles were left for the following calculation. Subsequently, bad particles were removed by using the ab-initio reconstruction in cryoSPARC and 3D-classifications in Relion. 294,021 particles were kept for the no-binning refinements and reconstruction, and the final reconstruction with resolutions of 3.23 Å was achieved (Supplementary Fig. 6 and Supplementary Table 1).

## Model building and refinement

The crystal structure of the DltB monomer (pdb code: 6buh) was used as an initial model. The highest resolution map of the DltB/m-AMSA complex at 3.23 Å was used for complete model building. The initial model was docked into the cryo-EM map and adjusted manually in coot[46]. The cryo-EM electron potential density map allowed us to build a model of the intact DltB protein as a tetramer, many detergent and lipid molecules as well as the inhibitor m-AMSA in each protomer. DltB apo and DltB/DltC complex models were modified based on the DltB/m-AMSA model. All structure refinements were carried out by PHENIX[47]. Structure figures were generated using PyMOL (http://www.pymol.org) and Chimera[48].

## Lipidomics analysis

Lipids extraction: Lipids from DltB overexpressing membrane or purified protein were extracted using methyl-tert-butyl ether (MTBE). 150 µL of methanol (MeOH) was added into 100 µL of the samples in a tube, followed by adding 500 µL of MTBE. The mixture was incubated for 1 h at room temperature in a shaker, then mixed thoroughly after adding 150 µL of MS-grade water. Centrifuge for 10 min at 15,000 × g

to separate the different phases, and move the upper phase containing MTBE to a new tube for evaporation.

LC-MS/MS analysis: LC-MS/MS analysis was performed on a Thermo Fisher Scientific Ultimate 3000 LC system equipped with an TripleTOF5600+ mass spectrometer (AB SCIEXTM). The LC system was comprised of an Diamonsil 5 μm C18(2) (4.6 mm × 150 mm) column. The mobile phase consisted of solvent A (0.1% HCOOH + 5 mM $HCOONH_4$ + 40% $H_2O$ + 60% ACN) and solvent B (0.1% HCOOH + 5 mM $HCOONH_4$ + 10% ACN + 90% isopropyl alcohol) were run with a gradient elution protocol (0–1 min, 20% B; 1–11 min, 20-100% B; 11–20 min, 100% B; 20.1–30 min, 20% B). The flow rate of the mobile phase was 0.3 mL·min$^{-1}$.

Detection was performed using Electrospray Ionization (ESI) in both positive and negative ion modes. The ESI source conditions were set as follows: Ion Source Gas1 (Gas 1): 50, Ion Source Gas2 (Gas 2): 50, Curtain Gas (CUR): 30, Source Temperature: 400 °C (positive ion mode) and 400 °C (negative ion mode), Ion Spray Voltage Floating (ISVF): 4500 V (positive ion mode) and 4000 V (negative ion mode), TOF MS scan range: 100-1500 Da, TOF MS scan accumulation time: 0.25 s, Declustering Potential (DP): ±80 V.

Identification of lipids: The raw data obtained from LC-MS was converted to ABF format using the Analysis Base File Converter software. The ABF format files were imported into MS-DIAL 4.70 software (MS-DIAL: data independent MS/MS deconvolution for comprehensive metabolome analysis. (Nature Methods, 12, 523–526, 2015)) for preprocessing. This included peak extraction, denoising, deconvolution, peak alignment, and exporting the data as a three-dimensional CSV format matrix (raw data matrix). The extracted peak information was compared with the database, and a comprehensive search of the LipidMap database was performed.

### Survival assays

DltB was knocked out in the *Bacillus Subtilis* 168 strain via homologous recombination using a modified pDG1730 integration vector. The *dltB* gene was replaced by a gene sequence encoding spectinomycin resistance, and the knockout clones were then selected on an LB plate containing 50 μg·mL$^{-1}$ spectinomycin. The *dltB* WT or mutant genes were cloned into a pHT expression vector, which was then transformed into the DltB knockout *B. Subtilis* 168 strain using the natural competence method. *B. Subtilis* 168 strains containing various DltB mutations were grown in 2 mL LB (supplemented with the appropriate antibiotics when needed) overnight at 37 °C. All cultures were adjusted to an OD600 of 0.25 and then serially diluted in LB broth with tenfold dilutions. For each strain, 5 μL of each dilution was plated onto LB plates with appropriate antibiotics, with or without 3 μg m$^{-1}$ of lysozyme (Sigma), and incubated at 37 °C overnight.

### Reporting summary

Further information on research design is available in the Nature Portfolio Reporting Summary linked to this article.

## Data availability

The atomic coordinates generated in this study have been deposited in the Protein Data Bank under the accession numbers 8JES (apo), 8JF2 (DltB/C complex), and 8JEM (DltB/m-AMSA complex). The cryo-EM electron potential density maps generated in this study have been deposited in the Electron Microscopy Data Bank (EMDB) under the accession numbers EMD-36194 (apo), EMD-36207 (DltB/C comples, and EMD-36192 (DltB/m-AMSA complex). The model of the DltB monomer (6BUH) was obtained from the PDB data bank. Uncropped scans of blots and gels in Figs. 1b-d, and 4c are supplied as Source Data. These presented in Supplementary Figs. are provided at the end of the Supplementary Information file. Lipidomics raw data generated in this study have been uploaded in MetaboLights under accession number MTBLS9864, and the processed data are provided in the Supplementary Information file. Source data are provided as a Source Data file. All other data are available from the corresponding authors upon request. Source data are provided with this paper.

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

## Acknowledgements
This work is supported by the startup grants from Renmin Hospital of Wuhan University (to P.Z.) and from Kobilka Institute of Innovative Drug Discovery in the Chinese University of Hongkong (Shenzhen) (to Z.L.).

## Author contributions
P.Z. initialized the project. Both authors (P.Z. and Z.L.) performed experimental studies and data interpretation. Both authors (P.Z. and Z.L.) were involved in writing the paper.

## Competing interests
The authors declare no competing interests.
