## [Peer Review File · Nature Communications]

Structural Insights into the Transporting and Catalyzing Mechanism of DltB in LTA D-alanylationReviewers' Comments:

Reviewer #1:

Remarks to the Author:

This manuscript increases knowledge on the structure and function of the lipoteichoic acid alanylation machinery of Gram-positive bacteria including major pathogens such as *S. aureus* significantly. I am not a structural biologist, my comments refer more to the overall presentation of the study.

- The authors do not address the recent biochemical study by Suzan Walker's lab, which presented important additional findings on the Dlt system (Schultz et al, 2023, Nat Microbiol), which was probably published after this manuscript was completed. The two studies appear to be largely complementary. Discussing their findings in the light of the Walker study would substantially improve the implications of the manuscript.
- The authors should indicate in Abstract and Results from which bacterial species the proteins they analyze were derived.
- Many abbreviations should be explained, e.g. in the Abstract (TMH, AMSA), Introductio (SOAT, ACAT, DGATI, PORCN,...).
- It is interesting to note that lipid molecules were found associated with DltB. However, the point remained a bit confusing to me. Why did the authors model phosphatidylglycerole into the structure although the D-Ala group is transferred to LTA polymers? Do the authors propose that DltB binds several, different lipid molecules? At another position in the Results the authors interpret the presence of diacylglycerol, again, not a substrate for D-alanylation.

Reviewer #2:

Remarks to the Author:

In this study, the authors utilized cryo-electron microscopy (cryo-EM) to investigate the structures of the DltB tetramer, an MBOAT protein crucial for cell wall biogenesis in Gram-positive bacteria, in three distinct forms. While the obtained structures showed similarities to those previously determined through crystallography, they also provided novel insights into the assembly of higher-order complexes. Despite the valuable findings, the current manuscript lacks essential functional analyses regarding the oligomeric state of DltB and its implications on its function *in vivo* and *in vitro*. Such information is vital to support the claims made by the authors and to enhance the overall impact of this study. Specifically, the authors claimed that DltB functions as tetramers, but they have not provided substantial evidence to support this assertion. Although the structure suggested specific residues crucial for tetramerization, their validity remains unverified by biochemical analysis. The inability of two mutants to express themselves individually does not sufficiently demonstrate the protein's tetrameric state.

It appears that this study is not ready for publication in Nature Communications, so it requires revision; specifically, the more functional analysis should be included. The basic logic of the writing is easy to follow; however, the text contains some careless errors that should not have occurred. The entire article is written in the simple present tense, so it is recommended to use the past tense when describing specific experiments conducted in the past.

Specific Comments:

Page 2: Should "serving" be replaced by "serves" to complete the sentence? "LTA" needs to be defined the first time it appears, not later in the introduction.

Page 4: Ref. 25 and Ref. 26 are not direct evidence supporting the authors' claims. Please find and refer to the original studies.

Page 6: "The tetramers aggregate or dissociate from batch to batch" seems to be caused by inconsistency between preparations.

Page 8: "Several critical lipids" What makes these lipids "critical"? Any evidence? It should be "two long helices". Use "C4 symmetry" or "4-fold symmetry" instead of "C4 fold".

Page 10: The description of MoIA-MoID and interfaces is required for a figure here. See comments for Figure 2.

Page 12: The fact that the two mutants did not express it does not support the claim.

Page 18: I do not find any grounds for the statement "reveals its native tetramer organization on membrane". It should be "donor", not "doner".

Figure 2

A cartoon diagram of the tetramer in top view, with the protomers and helices labeled. It will be helpful to illustrate the oligomeric organization and the different interfaces. Panel b is poorly constructed and should be replaced.

Reviewer #3:

None

REVIEWER COMMENTS

Reviewer #1 (Remarks to the Author):

This manuscript increases knowledge on the structure and function of the lipoteichoic acid alanylation machinery of Gram-positive bacteria including major pathogens such as *S. aureus* significantly. I am not a structural biologist, my comments refer more to the overall presentation of the study.

Response to reviewer 1: We appreciate the reviewer for his/her positive evaluations of our study. In the revised version, we have thoroughly considered the reviewer's criticisms and advices, suggestions, resulting in significant improvements to our manuscript.

- The authors do not address the recent biochemical study by Suzan Walker's lab, which presented important additional findings on the Dlt system (Schultz et al, 2023, Nat Microbiol), which was probably published after this manuscript was completed. The two studies appear to be largely complementary. Discussing their findings in the light of the Walker study would substantially improve the implications of the manuscript.

Response: Thanks for the reviewer's great advices, which helps to improve the completeness of the story and make our structures more intriguing. Dr. Walker's paper became available online during the review process of this submission. In this revised version, we have now discussed their findings and integrated their insights in our manuscript, both in the results session (DltB Inhibition by m-AMSA from Extracellular Surface) and in the discussion session (New Ref. 11). Both of our findings are largely complementary, particularly in the case of the LTA D-alanylation process. The difference lies in the predicted model of the DltBDX complex, which is not compatible with our DltB tetramer model, as the DltDX is located on the DltB tetramer interface (see the figure below). We believe the DltB tetramer interface may tend to be predicted as an interface for complexes; however, DltDX may actually bind somewhere else. This will require further structural studies.

- The authors should indicate in Abstract and Results from which bacterial species the proteins they analyze were derived.

Response: The DltB and DltC genes in this study were from *Streptococcus thermophilus*. In the previous version, we only had this information in the Methods session; now we also include it in the Abstract and Results sessions.

- Many abbreviations should be explained, e.g. in the Abstract (TMH, AMSA), Introduction (SOAT, ACAT, DGATI, PORCN,...).

Response: We have included the full names for these abbreviations when they first appear, and then use the abbreviations.

- It is interesting to note that lipid molecules were found associated with DltB. However, the point remained a bit confusing to me. Why did the authors model phosphatidylglycerol into the structure although the D-Ala group is transferred to LTA polymers? Do the authors propose that DltB binds several, different lipid molecules? At another position in the Results the authors interpret the presence of diacylglycerol, again, not a substrate for D-alanylation.

Response: The hydrophobic cleft beside the active site was defined as the substrate binding site, which binds the Lipid-anchored LTA, the substrate of DltB. Since PG has been suggested as an intermediate LTA anchor during LTA biosynthesis, and the shape of electron density fits a PG molecule fairly well, especially the glycerophosphate head group, we built PG in the substrate binding site. Additionally, we performed a lipidomics analysis of different samples during DltB purification, which revealed a significant PG accumulation on both DltB tetramer and monomer. About half of the lipids in the purified DltB tetramer protein are PG. Moreover, in the monomer fractions, more than 50% lipids are PG (see New Figure 3e), this confirms our model.

Yes, we identified several different lipid molecules in the structure. On the dimer interfaces, the density fits DAG molecules very well. Since it is on the interface, it does not serve as a substrate for D-alanylation. Furthermore, consistent with the lipidomics data, there is a significant amount of DAG in the purified DltB tetramer and monomer.

Indeed, there are also more lipids identified in the purified DltB sample according to the lipidomics data. We did model a few lipids (PG) in the detergent matrix (see Extended Data Table 1), however, due to the limited resolution, it is hard to accurately assign these molecule identities. They could be some other lipids identified by lipidomics analysis, such as MGDG, DGDG, etc.

Reviewer #2 (Remarks to the Author):

In this study, the authors utilized cryo-electron microscopy (cryo-EM) to investigate the structures of the DltB tetramer, an MBOAT protein crucial for cell wall biogenesis in Gram-positive bacteria, in three distinct forms. While the obtained structures showed similarities to those previously determined through crystallography, they also provided novel insights into the assembly of higher-

order complexes. Despite the valuable findings, the current manuscript lacks essential functional analyses regarding the oligomeric state of DltB and its implications on its function in vivo and in vitro. Such information is vital to support the claims made by the authors and to enhance the overall impact of this study. Specifically, the authors claimed that DltB functions as tetramers, but they have not provided substantial evidence to support this assertion. Although the structure suggested specific residues crucial for tetramerization, their validity remains unverified by biochemical analysis. The inability of two mutants to express themselves individually does not sufficiently demonstrate the protein's tetrameric state.

Response to reviewer 2: We thank the reviewer for the constructive comments on our manuscript. In order to substantiate the claim that DltB functions as tetramers, we have demonstrated that most DltB molecules form tetramers on the membrane using gel filtration experiment on the mildly solubilized DltB-overexpressing membrane (see **New Figure 1d**). We hope this new experiment with positive results will address the reviewer's concerns and enhance the quality of this manuscript.

It appears that this study is not ready for publication in Nature Communications, so it requires revision; specifically, the more functional analysis should be included. The basic logic of the writing is easy to follow; however, the text contains some careless errors that should not have occurred. The entire article is written in the simple present tense, so it is recommended to use the past tense when describing specific experiments conducted in the past.

Response: In addition to including the result of DltB tetramer organization on the membrane, we have also included the mass spectrometry data on the lipid composition of DltB samples during purification (see **New Figure 3e**), which suggests PG is specifically accumulated in the purified DltB tetramer and monomer. Thus, we provided more direct evidence to support our structural model, specifically regarding the lipids assignment, which strongly supports a PG molecule binding in the substrate binding site. Thus we demonstrated that PG serves as the LTA carrier bound to DltB for D-alanylation.

We have addressed several mistakes as suggested below, and have also changed the article to past tense.

Specific Comments:

Page 2: Should “serving” be replaced by “serves” to complete the sentence? “LTA” needs to be defined the first time it appears, not later in the introduction.

Response: Sorry for the carelessness, we have corrected these mistakes.

Page 4: Ref. 25 and Ref. 26 are not direct evidence supporting the authors' claims. Please find and refer to the original studies.

Response: We have double checked these two references. Ref. 25 proposed that DltB provides a putative channel for D-alanyl-Dcp. We have added the original study (**Ref. 9**) which showed sequence similarity with a variety of transport proteins based on a blast search. No other direct experimental evidence supports DltB serving as a transporter so far.

We have replaced Ref. 26 with a review paper from 1994 which proposed that LTA is synthesized on the outer membrane. Additionally, we have included two papers from 2007 which firstly described the LTA synthase LtaS and identified LtaA which facilitates the LTA precursor lipid from the inner leaflet to the outer leaflet.

Page 6: “The tetramers aggregate or dissociate from batch to batch” seems to be caused by inconsistency between preparations.

Response: Yes we believe that’s the case, and we have rephased this part to be clearer.

Page 8: “Several critical lipids” What makes these lipids “critical”? Any evidence? It should be “two long helices”. Use “C4 symmetry” or “4-fold symmetry” instead of “C4 fold”.

Response: We have defined the critical lipids as those found in the substrate binding site or on the tetramer interfaces. Additionally, we have provided MS data to support our modeling (see New Figure 3e).

Sorry for the carelessness in typing, we have corrected these mistakes.

Page 10: The description of MolA-MolD and interfaces is required for a figure here. See comments for Figure 2.

Response: We replaced panel b in Figure 2 with a new figure demonstrating the DltB tetramer organization, with the protomers and helices labeled. This highlights the MolA-B and MolB-C interfaces as well.

Page 12: The fact that the two mutants did not express it does not support the claim.

Response: It’s unfortunate that the two mutants on the intratetramer interface did not express. Since they are on the transmembrane surface of the protein, the mutation may affect the protein’s stability or folding, leading to rapid degradation and rendering them undetectable. We agree that the original claim “indicates the tetramer is critical for the stability of DltB protein” is not appropriate. We have changed to “This suggests that the intratetramer interface is critical for the stability of DltB protein”.

Page 18: I do not find any grounds for the statement “reveals its native tetramer organization on membrane”. It should be “donor”, not “doner”.

Response: As we have added more evidence to show most DltB molecules form tetramers on the membrane (see New Figure 1d), we rephrase it here: “Our biochemical analysis and cryo-EM structure of DltB revealed its native tetramer organization on membrane”.

We have corrected these mistakes.

Figure 2

A cartoon diagram of the tetramer in top view, with the protomers and helices labeled. It will be helpful to illustrate the oligomeric organization and the different interfaces. Panel b is poorly constructed and should be replaced.

Response: We have replaced it with new Figure 2b as shown below:

Reviewers' Comments:

Reviewer #2:

Remarks to the Author:

I think the authors have partially addressed my concerns. However, it is still not clear which residues/lipids are important for tetramerization and how the oligomeric state affects the function. I prefer the authors answer at least one of these questions before this manuscript is accepted.

REVIEWER COMMENTS

Reviewer #2 (Remarks to the Author):

I think the authors have partially addressed my concerns. However, it is still not clear which residues/lipids are important for tetramerization and how the oligomeric state affects the function. I prefer the authors answer at least one of these questions before this manuscript is accepted.

Response: We appreciate the insistence on gathering more evidence related to the tetramerization and its importance for DltB function. To address the concern, we had made the following additions to the manuscript:

1. Generation of Four Single Mutations on the Tetramer Interface: Previously, we generated two double mutations (I42R/L46R and M199A/L200R) on the tetramer interface, but no expression of DltB was detected in *E. coli* C41 cells. In this revision, we have instead generated four single mutations (I42R, L46R, M199A, and L200R) on the tetramer interface. All these mutants expressed well in *E. coli* C41 cells, similar to the DltB wild-type. We assessed their oligomerization status on the cell membrane using a mild detergent solubilization condition. Without surprise, all these mutations impaired the tetramer organization compared to the WT DltB protein (new Extended Data Fig.8).

2. Lysozyme-Sensitivity Survival Assay in *Bacillus subtilis* 168: Additionally, we generated a *dltB* knockout strain of gram positive bacteria *B. subtilis* 168, reconstituted it with DltB WT or the four mutants, and conducted a lysozyme-sensitivity survival assay to evaluate the role of tetramer in LTA D-alanylation. Our results revealed that while the WT DltB protein rescued the lysozyme vulnerability, the tetramer interface mutants failed to do so (new Fig. 4b). We observed that the expression level of DltB protein in *B. subtilis* 168 was much lower than that in *E. coli* C41 cells (new Fig. 4c).

As for the lipids crucial for DltB function, we have identified phosphatidylglycerol (PG) lipids from purified DltB samples (both monomer and tetramer) using mass spectrometry (Fig. 3e). Subsequently, we have incorporated a PG molecule into the model (Fig. 3c,d). Given its direct connection to the active site of DltB, we propose that this lipid serves as the carrier for LTA. Regarding critical lipids for tetramerization, by combining the lipidomic data and cryo-EM electron density map, we have positioned a DAG molecule at the tetramer interface. However, further investigations using other biochemical or biophysical methods are warranted to confirm this model.

Taken together, our findings suggest that DltB tetramers indeed exist on the cell membrane to perform their physiological functions in LTA D-alanylation. These tetramer likely serves as a scaffold for conformational change, facilitating the transfer of the D-alanyl group from DltC to the active site and LTA substrate on the outside of DltB. These additions enhance the significance and interest of the manuscript.

Reviewers' Comments:

Reviewer #1:

Remarks to the Author:

The authors performed several additional experiments that support and extend the conclusions. I could not find the new figures 3e and 4c in the revised manuscript though. Did I miss something or do these figures need to be added?

Reviewer #2:

Remarks to the Author:

The authors have addressed any remaining concerns.

REVIEWERS' COMMENTS

Reviewer #1 (Remarks to the Author):

The authors performed several additional experiments that support and extend the conclusions. I could not find the new figures 3e and 4c in the revised manuscript though. Did I miss something or do these figures need to be added?

Response: The figure 3e and 4c were included in the last revised PDF manuscript. They are uploaded as separate figure files for this final revision. I have also attached a copy here for your reference.

Fig.3

Fig.4

a

b

c

Reviewer #2 (Remarks to the Author):

The authors have addressed any remaining concerns.

Response: Thank you!